# Thermally Conductive and Electrically Insulating Epoxy Composites Filled with Network-like Alumina In Situ Coated Graphene

**DOI:** 10.3390/nano13152243

**Published:** 2023-08-03

**Authors:** Ruicong Lv, Haichang Guo, Lei Kang, Akbar Bashir, Liucheng Ren, Hongyu Niu, Shulin Bai

**Affiliations:** 1School of Materials Science and Engineering, HEDPS/Center for Applied Physics and Technology, Peking University, Beijing 100871, China; 1801111671@pku.edu.cn (R.L.); haichang@pku.edu.cn (H.G.);; 2Peking University Nanchang Innovation Institute, 14#1–2 Floor, High-Level Talent Industrial Park, High-Tech District, Nanchang 330000, China

**Keywords:** graphene, alumina, epoxy composite, thermal conductivity, insulation

## Abstract

With the rapid development of the electronics industry, there is a growing demand for packaging materials that possess both high thermal conductivity (TC) and low electrical conductivity (EC). However, traditional insulating fillers such as boron nitride, aluminum nitride, and alumina (Al_2_O_3_) have relatively low intrinsic TC. When graphene, which exhibits both superhigh TC and EC, is used as a filler to fill epoxy resin, the TC of blends can be much higher than that of blends containing more traditional fillers. However, the high EC of graphene limits its application in cases where electrical insulation is required. To address this challenge, a method for coating graphene sheets with an in situ grown Al_2_O_3_ layer has been proposed for the fabrication of epoxy-based composites with both high TC and low EC. In the presence of a cationic surfactant, a dense Al_2_O_3_ layer with a network structure can be formed on the surface of graphene sheets. When the total content of Al_2_O_3_ and graphene mixed filler reached 30 wt%, the TC of the epoxy composite reached 0.97 W m^−1^ K^−1^, while the EC remained above 10^11^ Ω·cm. Finite element simulations accurately predicted TC and EC values in accordance with experimental results. This material, with its combination of high TC and good insulation properties, exhibits excellent potential for microelectronic packaging applications.

## 1. Introduction

The high power density and frequency of modern electronic devices require effective heat dissipation in operation. This has stimulated the development of thermally conductive and electrically insulating materials. Traditional epoxy resin has low thermal conductivity (TC) and cannot meet the needs for practical application in thermal management [1]. Filling the polymer matrix with particles having higher TC may have a significant effect on TC. Common fillers include alumina (Al_2_O_3_) [2,3], boron nitride (BN) [4,5], graphite flakes [6,7], and graphene [8,9,10,11]. Al_2_O_3_ and BN are insulating fillers, but their intrinsic TC is not as high as that of graphene. When graphene is used to fill epoxy resin, due to its high intrinsic EC, the low resistivity of the composites cannot meet the requirements for packaging materials [12]. Therefore, it is necessary to conduct a surface insulation treatment for graphene to meet the market demand for simultaneously high TC and good insulation properties.

The surface treatment of graphene can be divided into minor molecule graft modification and surface deposition coating. The former requires grafting smaller molecular chains onto the ends or surface of graphene to reduce interface thermal resistance and improve compatibility between graphene and the matrix [13,14,15,16]. This modification also helps to enhance the overall thermal conductivity of the composites [17]. This type of treatment method can generally be divided into covalent and non-covalent modification [18]. The latter involves the growth of a coating layer on the surface of graphene [19,20,21,22]. Since graphene itself has high EC, it will reduce the insulation performance of composites when used as a filler. Appropriate growth of the insulative coating layer can improve the surface properties of graphene and enhance the insulating performance of the composites.

Recently, the surface coating of graphene with different insulating layers has been achieved, effectively increasing the intrinsic resistivity of graphene. Using a sol-gel method, a silica (SiO_2_) coating on graphene oxide (GO) was prepared [23]. The TC of the GO–SiO_2_ sandwich structure was 0.322 W m^−1^ K^−1^ at 1 wt% GO–SiO_2_ total loading, and the resistivity remained at 2.96 × 10^11^ Ω cm. In another example, tetraethyl orthosilicate (TEOS) was used as a precursor pre-grafted with 3-aminopropyl triethoxysilane (APTES) to generate nano-SiO_2_ in situ to coat graphene sheets and to enhance the compatibility between graphene and epoxy [24]. The epoxy blend prepared from this method had a TC of 0.297 W m^−1^ K^−1^ at 8 wt% loading. As a final example, Al_2_O_3_ deposited on a graphite sheet was used to prepare a thermally conductive and insulating phthalonitrile composite by hot pressing [25]. Electrical insulating properties were maintained, and the TC reached 0.6 W m^−1^ K^−1^ at a total loading of 20 wt%. In order to achieve both heat conduction and electrical insulation simultaneously, it is necessary to uniformly grow a coating layer with a specific structure on the surface of the fillers.

Here, a method for growing a network-like Al_2_O_3_ coating on the surface of graphene in the presence of a cationic surfactant has been developed. The coating obtained in this way is more compact, and the thermally conductive and electrically insulative performance is better than that of directly deposited coating. In maintaining bulk resistivity over 10^11^ Ω cm, the TC of the epoxy-based composite reached 0.97 W m^−1^ K^−1^ at 30 wt% loading. In addition, a finite element simulation was carried out to characterize the effect of a surface coating of graphene on the TC and EC of epoxy blends. The comparison between the simulation and tests results shows good agreement. Figure 1 shows the whole process for the composite preparation.

## 2. Experimental Section

### 2.1. Materials

Graphene sheets with a lateral size of 10 μm were purchased from Xiamen Knano Graphene Technology Co, Ltd., Xiamen, China. Aluminum sulfate octadecahydrate was supplied by Beijing Tongguang Fine Chemicals Company, China. NaOH was provided by Guangdong Xilong Scientific, China. Cetyltrimethylammonium bromide (CTAB) was purchased from Aladdin. Epoxy (E51), curing agent (593), and diluent (501) were purchased from Shanghai Lewei, China. Ethanol and distilled water were used on an instant basis.

### 2.2. Preparation of Al_2_O_3_@Graphene Powder

An amount of 10 g of aluminum sulfate octadecahydrate was dissolved in 40 mL of deionized water, followed by the addition of 1 g of graphene sheets and 0.01 g of CTAB. The suspension was then mixed in a defoaming vacuum centrifuge at 1500 rpm for 5 min. In parallel, the stoichiometric ratio of NaOH was dissolved in 80 mL of deionized water. The NaOH solution was added dropwise into the mixture at a constant temperature of 60 °C with continuous stirring. The addition was divided into four parts, with an interval of 30 min between each addition, and the stirring was continued for 2 h after the complete addition. The resulting suspension was centrifuged and washed three times with deionized water and ethanol. Subsequently, the obtained sample was freeze-dried for 12 h. Finally, the freeze-dried alumina–graphene sample was ground, placed in a crucible, and heated at 540 °C for 3 h to obtain the desired filler, referred to as Al_2_O_3_@Graphene–CTAB. Another set of samples was prepared using the same method without the addition of CTAB, referred to as Al_2_O_3_@Graphene–NCTAB.

### 2.3. Preparation of Al_2_O_3_@Graphene Epoxy Composites

The epoxy resin, curing agent, and diluent were mixed with the calculated amount according to the ratio of 100:25:15. Then, the specified amount of Al_2_O_3_@Graphene–CTAB was added and stirred in the defoaming vacuum mixer at 1500 rpm for 5 min. Finally, the mixture was poured into the poly tetra fluoroethylene (PTFE) mold and cured for 180 min at 120 °C to obtain composite samples.

### 2.4. Characterization

A scanning electron microscope (SEM, S-4800, HITACHI, Tokyo, Japan) was used to observe the morphology of the modified graphene and the fractured surface of the composites. Energy dispersive X-ray spectroscopy (EDX, Apollo XP, USA) was used to characterize the element distribution on the surface of the mixed fillers. The surface topography of the modified graphene was measured using transmission electron microscope in 100 kV (TEM, FEI Talos F200S, Carlsbad, CA, USA). Thermogravimetric analysis (TGA, Q600, TA Instrument, New Castle, DE, USA) was used to measure the weight loss of the sample during the heating process in the air environment at 20 °C/min. The crystal form was measured by powder X-ray diffraction from 5–80° (XRD, X-Pert3 Powder, Almelo, The Netherlands).

TC was calculated by the equation: *λ* = *α* × *C_p_* × *ρ*. *λ*, *α*, *C_p_*, and *ρ* represent TC, thermal diffusivity, specific heat capacity, and density, respectively. *α* was measured by a laser flash instrument (DXF-500, TA Instrument, New Castle, DE, USA) at room temperature on the sample of 12.7 mm in diameter. *C_p_* was obtained by differential scanning calorimetry (DSC, Q2000, TA Instrument, New Castle, DE, USA) ranging from −5 °C to 50 °C at the heat rate of 5 °C/min. *ρ* was measured by an analytical balance (XS204, Columbus, OH, USA) based on Archimedes’ principle. The electrical performance was characterized by a high resistance weak current tester (ST2643, Suzhou Jing Ge, Suzhou, China).

## 3. Results and Discussion

### 3.1. Characterization of Pristine Graphene and Al_2_O_3_@Graphene

In order to characterize the effect of the Al_2_O_3_ coating on the graphene surface morphology, SEM images were taken as shown in Figure 2. Figure 2a,d show the smooth surface of pure graphene at different magnifications. Figure 2b,e represent Al_2_O_3_ coated graphene in the presence of CTAB surfactants. It was found that a dense network cladding structure was formed on the graphene surface. The cladding samples without CTAB in Figure 2c,f show a rougher surface than that of pure graphene, but have no network-like structure. It is well known that the graphene surface has a negative Zeta potential, so the use of CTAB surfactant with a positive charge is beneficial to the directional deposition of Al_2_O_3_. With the slow addition of the alkaline solution, the newly generated aluminum hydroxide is more inclined to deposit on the graphene surface. In the absence of surfactants, aluminum hydroxide is more likely to grow on the existing crystal nuclei.

EDX detection was used to verify the aluminum distribution on the graphene surface after the treatment in the presence of CTAB. As is shown in Figure 3a–c, the density of the C element on the surface was significantly less than that of O and Al, confirming a dense Al_2_O_3_ layer on the surface. This can be seen more clearly on the total elements distribution observed in Figure 3d. Compared with the smooth graphene sheet, the uniformity of the Al_2_O_3_ layer can be observed from Figure 3e,f, which agrees well with the results obtained in the previous EDX.

Unlike with direct blending of different fillers, the respective filler content cannot be directly obtained in this way because Al_2_O_3_ is formed in situ. In order to have the exact proportions of different components, the TGA curve of the fillers was analyzed. As is shown in Figure 4a, pure graphene started to lose weight at 526 °C and lost 93.08 wt% at 737 °C. The graphene coated with Al_2_O_3_ started to lose weight at 549 °C and lost 24.56 wt% at 889 °C. This means that the graphene occupied about 25 wt% of the mixed filler. The higher the relative proportion of graphene in the mixed filler, the higher the TC of the powder, but also the higher the probability of forming an electrically conductive path inside the composites.

Furthermore, the moldability of the composites was significantly enhanced with the incorporation of the Al_2_O_3_ coating. For instance, when preparing a composite with 15 wt% pure graphene using vacuum defoaming, the epoxy resin faced challenges in fully infiltrating the graphene structure. However, with the presence of the Al_2_O_3_ coating, even at a higher loading of 20 wt% Al_2_O_3_@graphene, the mixture exhibited excellent fluidity and uniform dispersion during the fabrication process. This improved moldability is attributed to the optimized interaction between the Al_2_O_3_ coating and the epoxy resin, enabling better wetting and penetration of the filler materials.

XRD was used to further characterize the structure of the in situ coating layer as shown in Figure 4b. As a control group, pure Al_2_O_3_ powders obtained by precipitation sintering have the peaks of AlOOH and Al_2_O_3_ [26]. The decomposition of AlOOH is a very complicated process. Below 1000 °C, it is a state where multiple crystal forms coexist [27]. Thus, the pure Al_2_O_3_ sample was just used to make a comparison with the experimental group to characterize the change of peak position before and after coating. It is known from the previous TGA diagram that graphene will be thermally decomposed above 550 °C, so the heating temperature we chose was 540 °C. However, the aluminum hydroxide precipitated at this temperature cannot ultimately produce Al_2_O_3_, so the peak positions of AlOOH at 19°, 23°, and 66° can still be observed. The intrinsic peaks of graphene can be observed at 26.55° and 54.66° [28], which presents the stacking of multilayer graphene sheets. It can be seen from Figure 4b that the addition of surfactant CTAB did not make a significant difference in the XRD peak shape of the fillers, meaning that the existence of CTAB does not have a significant impact on the crystal form of Al_2_O_3_. Apparent differences in the morphology of the coating layer indicate that the impact is more on the generation site of aluminum hydroxide.

### 3.2. Morphology and Thermal Stability of Composites

Figure 5a,e display the cross-sectional SEM images of pure epoxy, exhibiting a very smooth surface. In contrast, Figure 5b–d,f–h showcase the cross-sectional SEM images of epoxy composites filled with 10, 20, and 30 wt% Al_2_O_3_@Graphene–CTAB, respectively. At lower magnification, it is evident that the graphene fillers were uniformly dispersed throughout the epoxy matrix, which can be attributed to the presence of the surface coating. At higher magnification, a dense Al_2_O_3_ layer can be observed on the surface of the graphene sheets. Notably, Figure 5f clearly illustrates the well-structured coating, effectively isolating the graphene sheets and forming an insulation system consisting of the graphene–Al_2_O_3_ layer epoxy matrix.

In order to characterize the actual proportion of components in the composites, the TGA curves in air condition were also studied. The TGA curve of pure epoxy resin in Figure 6 shows that the weight loss was 81.45% from 78 °C to 497 °C and 19.01% from 497 °C to 692 °C, respectively. It can also be seen from Figure 6 that the 30 wt% Al_2_O_3_@Graphene-filled epoxy composite had a thermal weight loss of 58.58% and 15.35% at 484 °C and 617 °C, respectively, which corresponds to the thermal weight loss of pure epoxy. In addition, there was a 7.16% thermal weight loss at 831 °C, which corresponds to the partial weight loss of graphene. As is shown in Figure 4a, the Al_2_O_3_-coated graphene started to lose thermal weight at 549 °C, indicating that there was also the decomposition of the filler before 617 °C. After heating to 1000 °C, the pure epoxy sample was almost completely burnt, while the 30 wt% Al_2_O_3_@Graphene-filled epoxy composite still had 18.76% residue. The quantity of the remaining white Al_2_O_3_ powder was consistent with the proportion of the input amount.

### 3.3. Thermal and Electrical Properties of Composites

In the design of thermally conductive and electrically insulative packaging materials, achieving optimal thermal conductivity (TC) and electrical resistivity (EC) is of utmost importance. The insulation performance should meet a minimum value of 10^9^ Ω·cm while maximizing the TC to ensure efficient heat dissipation. By increasing the graphene content in the composites, the TC increases while the volume resistivity gradually decreases. Therefore, it is crucial to carefully select the total filler content and coating ratio in order to achieve the desired properties.

The selection of an appropriate total filler content and coating ratio is critical in the design of composites for thermally conductive and electrically insulative packaging materials. Through careful optimization, it becomes possible to simultaneously achieve the desired TC and EC properties, ensuring that the materials meet the insulation performance requirements while maximizing thermal conductivity for efficient heat dissipation.

From Figure 7a, the TC of the composites increased approximately linearly as the filler content increased gradually. The TC of the pure epoxy resin was 0.2 W m^−1^ K^−1^, and the TC of the composite with 30 wt% filler reached 0.97 W m^−1^ K^−1^. Comparing the two sets of Al_2_O_3_@Graphene–CTAB and Al_2_O_3_@Graphene–NCTAB-filled epoxy composites, no significant difference of TC value can be noticed at low filler loading levels. Only at 30 wt% loading did the composite with CTAB show a significantly higher TC than that without CTAB. Al_2_O_3_ without surfactants in composites cannot form a net-like structure coating, as shown in Figure 2f. The uniform net-like Al_2_O_3_ coating structure formed under the action of CTAB is very beneficial to maintain the dispersion of Al_2_O_3_@Graphene, as well as the insulative performance of composites.

Figure 7b gives the volume resistivity of the composites as a function of the filler contents. It can be seen that the resistivity of the composites was always larger than 10^11^ Ω·cm, which means that they can meet the requirements of insulation when applied. Below 20 wt% filler, the resistivity remained almost constant, indicating that the presence of the Al_2_O_3_ coating layer can well cover the electrically conductive graphene sheets. As the filler content increased further, even though the resistivity decreased rapidly, it was still two orders of magnitude higher than the critical value of 10^9^ Ω·cm at the 30 wt% loading. 

## 4. FEM Simulation of Thermal and Electrical Performance

In order to further understand the mechanisms of the thermal and electrical transfer, the finite element method (FEM) was used to simulate the TC and EC of Al_2_O_3_@Graphene-filled epoxy composites. On the microscopic scale, a representative volume element (RVE) was constructed to describe the distribution of the fillers. The steady-state method was employed to establish a local microscopic heat and electricity transfer model and to homogenize and evaluate the effective TC and EC on the macro scale [29,30,31].

The finite element modeling used in this study is based on Ansys APDL. As a two-dimensional filler with high aspect ratio, graphene can be geometrically modeled with a rectangle. In a cubic RVE of a given size, the total number of two-dimensional rectangles is fixed. The relative size of the rectangle is adjusted to fit the composites with the specified filler content. It is equivalent to selecting a cube area of a specified volume according to requirements in a uniform system and studying this area separately to characterize the overall macroscopic TC and EC.

In order to describe the ternary composite system filled with Al_2_O_3_-coated graphene, three materials were defined as epoxy resin (matrix), graphene (filler), and Al_2_O_3_ (interphase). There was no mutual contact between the fillers. In order to know the influence of the Al_2_O_3_ coating on the TC and EC of the graphene-filled epoxy composites, an interphase with a specified mass fraction was added to the surface of the rectangle. According to the TGA data in Figure 4a, the mass fraction of graphene in the Al_2_O_3_@Graphene filler accounted for about 25%, so a 75% mass fraction interphase existed on the original surface of the two-dimensional disc. Table 1 lists all the parameters of the materials introduced in the finite element simulation.

The thermal conductivity and electrical conductivity of graphene differ significantly from that of the matrix. Therefore, when assigning values in software, if the electrical conductivity of graphene is assigned as 10^6^ Ω cm, it would result in an excessive difference in magnitude that prevents accurate calculations. Due to the high resistivity of the alumina coating on the graphene surface, the values for graphene were assigned according to the data in Table 1 to facilitate the calculations.

### 4.1. Simulation of Heat Transfer

A steady-state heat transfer model was established by applying a temperature field to the left (high temperature of 323 °C) and right (low temperature of 298 °C) boundaries as shown in Figure 8. Three models of filler distribution were generated randomly for 10 wt% and 20 wt% Al_2_O_3_@Graphene–CTAB-filled epoxy composites. Heat transfer was studied in PLANE55. In order to obtain a random distribution model, x-y coordinates of the fillers and the angle formed by the filler length direction with the heat flow direction were used to describe the distribution state of the fillers. It can be seen that the position and orientation of fillers had a significant influence on the homogeneity of the temperature field. The location with fillers oriented in the heat transfer direction had a more protruding temperature field, meaning more rapid heat conduction due to the high TC of the fillers. The more random the filler distribution was, the more homogeneous the temperature field was.

Figure 9 shows the comparison of TC obtained by simulation and tests. The simulation results were obtained by extracting the X-direction heat flow of each node in the heat flow field and calculating it according to Fourier’s law after averaging. It can be seen from Figure 9 that the simulation results were very close to the experimental results.

### 4.2. Simulation of Electric Conduction

In order to describe the electrically conductive behavior of the filler inside the matrix, an electrical model was also established and the voltage field was simulated by Ansys APDL. The filler was randomly generated in the same way and the voltage field was calculated in PLANE230. For the same filler distribution model, the change of elements type can be used to calculate different physical fields. The steady-state method is also used to simulate the electrical field. Voltages of 5 V and 0 V were applied to the left and right sides of the model, and the average current density was calculated on the node to get the average resistivity of the whole.

Figure 10 shows the steady-state temperature field and voltage field distribution under the same model. Graphene, as a highly conductive material, has high TC and EC, but Al_2_O_3_ coating has simultaneously a higher TC than the matrix and a resistivity of up to 10^13^ Ω m, which makes the heat transfer and electrical conducting behavior different. It can be seen from Figure 10a that the temperature field was more uniform in the region where filler distribution was randomly distributed, indicating that the filler had a significant effect on heat transfer. However, the voltage dropping in the dense area of filler distribution is very obvious, which indicates that the coating layer blocked the transmission of the current, and had an insulation effect.

Figure 11 represents the comparison of electrical resistivity obtained by simulation and tests. On the whole, the decreasing trend of electrical resistivity with the increase of filler loading was similar for the simulation and the tests. The electrical resistivity decreased more quickly in the simulation than in the tests when the filler loading was below 20 wt%. The percolation effect of electrical resistivity seemed to appear for the tested results at 20 wt% filler loading. This is because in the actual conduction behavior, a conduction path will be formed, and the resistivity will decrease significantly when the filler loading increases to a threshold value. However, there is no threshold phenomenon in the simulation results, which depends on the model established.

## 5. Conclusions

In summary, graphene sheets were coated with an in situ formed Al_2_O_3_ layer and then added into epoxy to prepare thermally conductive and electrically insulative composites. Under the control of the cationic surfactant CTAB and by mixing aluminum sulfate octadecahydrate, CTAB, and NaOH with graphene sheets following a certain procedure, a net-like form of Al_2_O_3_ was formed in situ and covered the surface of the graphene sheets, which changed the graphene from conductive to insulating. At 30 wt% mixed fillers, the TC of the epoxy composite reached 0.97 W m^−1^ K^−1^, while the electrical resistivity remained above 10^11^ Ω·cm. Compared with pure graphene filling, this dense mesh Al_2_O_3_ coating can ensure the electric insulation of graphene-filled epoxy composites and so meets special requirements for both high thermal conductivity and electrical insulation in practical application. A finite element simulation was performed by introducing an Al_2_O_3_ coating on the graphene. A very good agreement between the simulated and experimental TC was obtained, and the dependance of electrical resistivity on the filler loading was similar for the simulation and the tests. This work provides a guiding approach for the insulation of electrically conductive graphene and a new design method and possible industrial application of thermoset composites having both high TC and good insulation properties.

## Figures and Tables

**Figure 1 nanomaterials-13-02243-f001:**
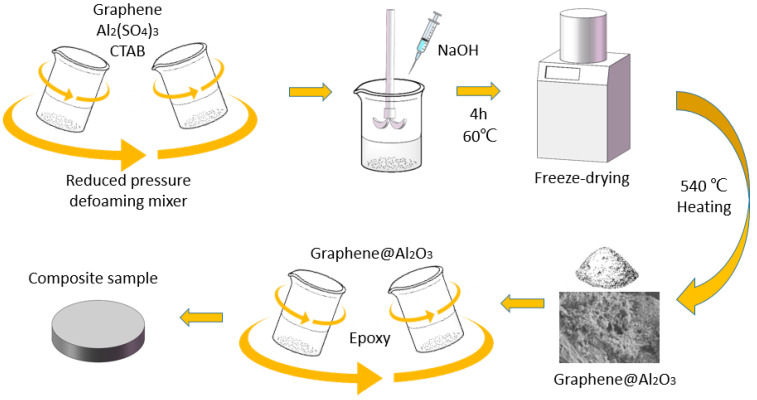
Labeled schematic illustration for the synthesis of Al_2_O_3_@Graphene-filled epoxy composites.

**Figure 2 nanomaterials-13-02243-f002:**
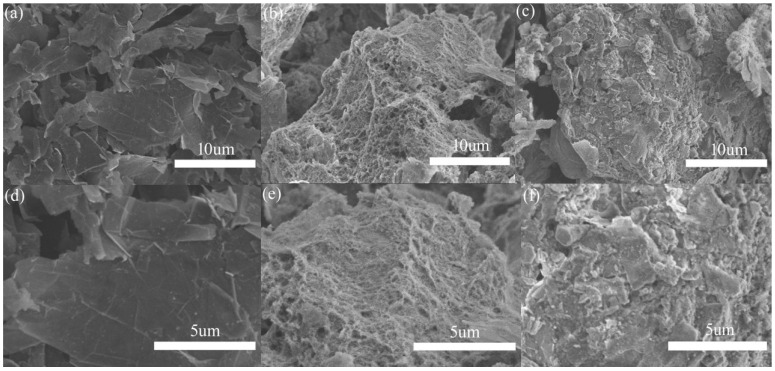
SEM micrographs of (**a**,**d**) pure graphene, (**b**,**e**) Al_2_O_3_@Graphene–CTAB, and (**c**,**f**) Al_2_O_3_@Graphene–NCTAB.

**Figure 3 nanomaterials-13-02243-f003:**
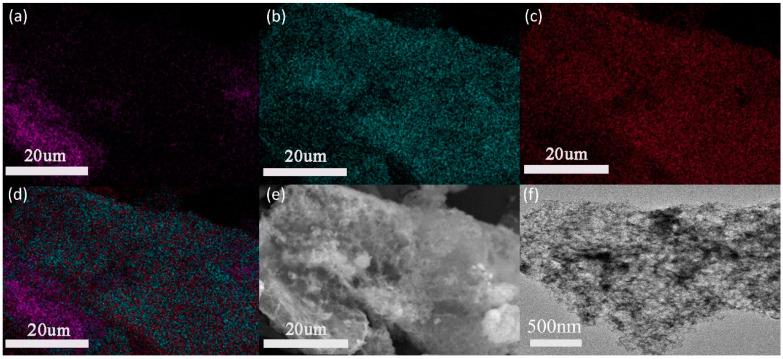
Energy-dispersive X-ray spectroscopy (EDS) of Al_2_O_3_@Graphene–CTAB: (**a**) C element; (**b**) O element; (**c**) Al element; (**d**) total elements composition; and (**e**) SEM and (**f**) TEM pictures of Al_2_O_3_@Graphene–CTAB.

**Figure 4 nanomaterials-13-02243-f004:**
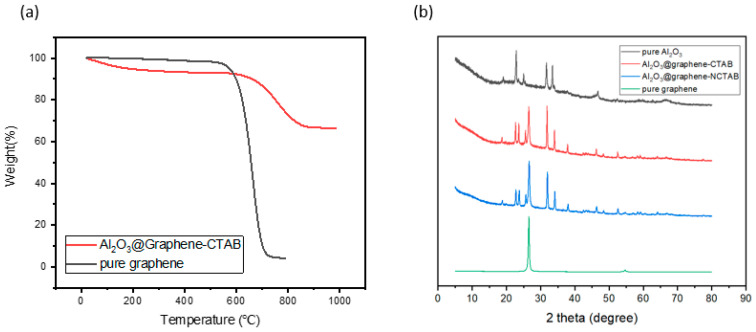
(**a**) TGA of pure graphene and Al_2_O_3_@Graphene–CTAB. (**b**) XRD of pure graphene, pure Al_2_O_3_, Al_2_O_3_@Graphene–CTAB, and Al_2_O_3_@Graphene–NCTAB.

**Figure 5 nanomaterials-13-02243-f005:**
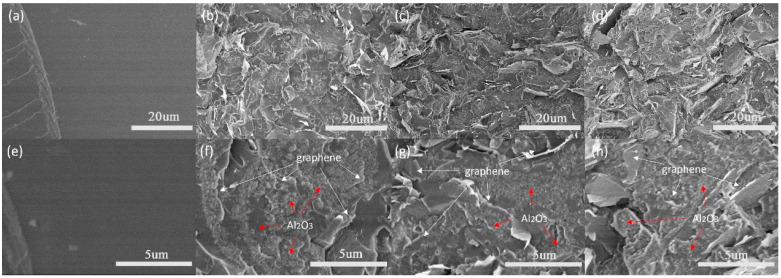
(**a**–**d**) Low and (**e**–**h**) high magnification of pure epoxy and 10, 20, and 30 wt% Al_2_O_3_@Graphene–CTAB-filled epoxy composites, respectively.

**Figure 6 nanomaterials-13-02243-f006:**
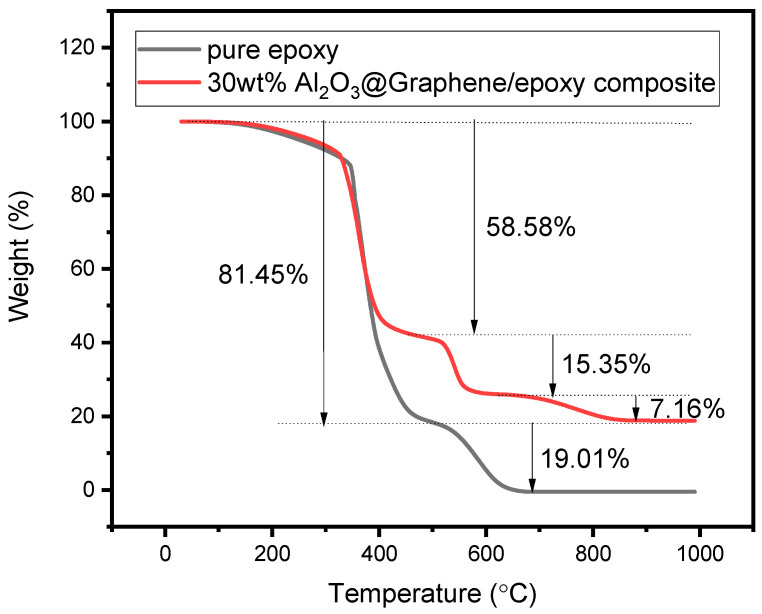
TGA of pure epoxy and 30 wt% Al_2_O_3_@Graphene–CTAB-filled epoxy composite.

**Figure 7 nanomaterials-13-02243-f007:**
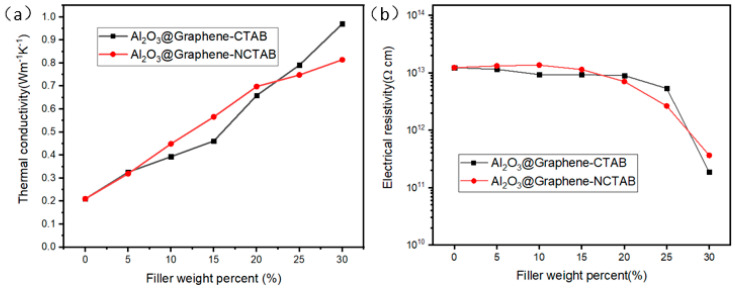
(**a**) TC and (**b**) electrical resistivity of Al_2_O_3_@Graphene-filled epoxy composites.

**Figure 8 nanomaterials-13-02243-f008:**
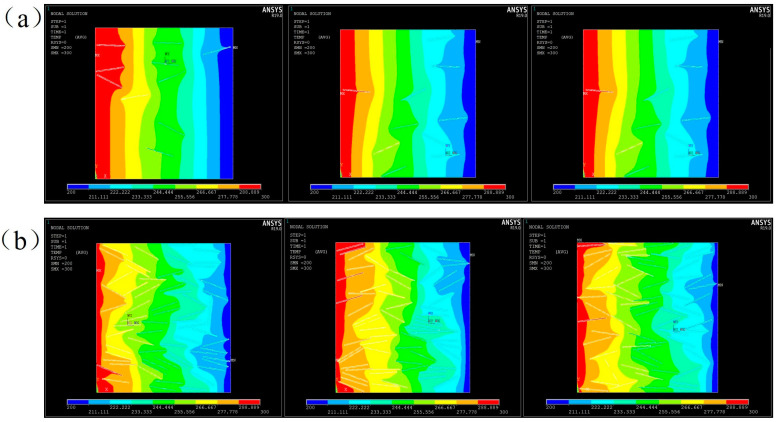
Temperature fields of three random models for (**a**) 10 wt% Al_2_O_3_@Graphene and (**b**) 20 wt% Al_2_O_3_@Graphene–CTAB-filled epoxy composites.

**Figure 9 nanomaterials-13-02243-f009:**
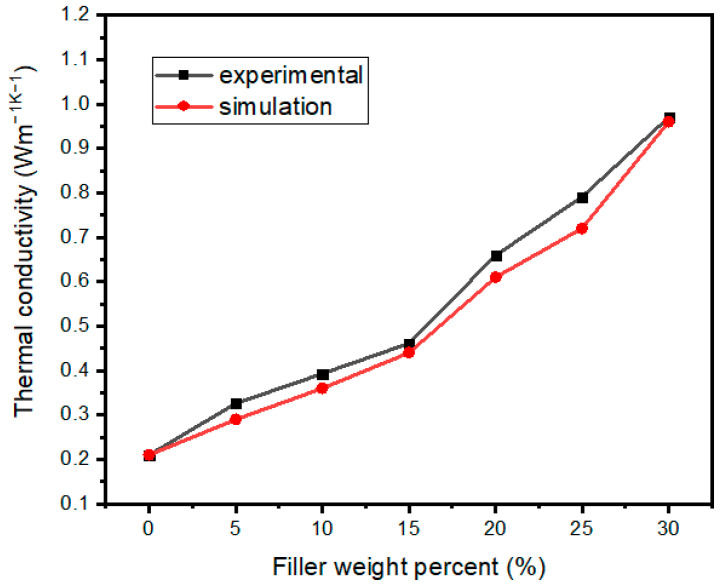
TC comparison among simulation and tests of Al_2_O_3_@Graphene–CTAB-filled epoxy composites.

**Figure 10 nanomaterials-13-02243-f010:**
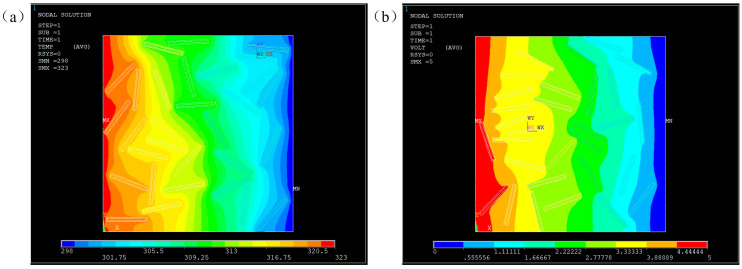
Simulated results of Al_2_O_3_@Graphene–CTAB-filled epoxy composites with (**a**) temperature field and (**b**) voltage field.

**Figure 11 nanomaterials-13-02243-f011:**
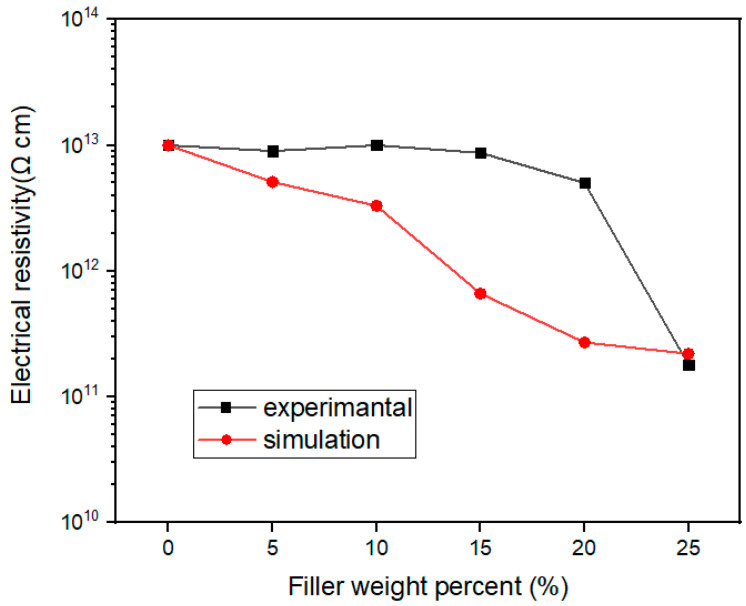
Electrical resistivity by simulation and tests of Al_2_O_3_@Graphene–CTAB-filled epoxy composites.

**Table 1 nanomaterials-13-02243-t001:** Parameters of materials used in finite element simulation.

	Density (kg m^−3^)	Specific Heat Capacity (J kg^−1^ K^−1^)	TC (W m^−1^ K^−1^)	Electrical Resistivity (Ω cm)
Epoxy	1200	1200	0.2	10^13^
Graphene	2200	700	through-plane: 5	10^−2^
			in-plane: 1800	
Al_2_O_3_	3500	850	30	10^14^

## Data Availability

All data is contained within the article.

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
