# Peer review of "Thermally Conductive and Electrically Insulating Epoxy Composites Filled with Network-like Alumina In Situ Coated Graphene"

_nanomaterials, 2023, doi:10.3390/nano13152243_

Round 1

Reviewer 1 Report

Th authors present a research on a network-like Al2O3 coating on the surface of graphene in the presence of a cationic surfactant and characterize the electrical and thermal properties. Development of electrically insulating and thermally conducting materials is important for the application to electronic devices. However, the originality of the manuscript is not high because some similar approaches have been reported. Other comments are below.

1. Graphene is one of the most conductive material. So authors should clarify the reason for the choice of graphene as the template for the electrically insulating composites.

2. Thermal conductivity of the materials is characterized in-plane and through the thickness direction. Which direction of the TC of the composite was measured? And why the authors measured on one direction?

3. Very similar paper was reported on the thermally conductive and electrically insulating epoxy composites with graphene/Al2O3. What is the differences and improvement of the manuscript.

4.  I suggest that the authors add the potential applications of the resultant composites and preliminary data.

Reviewer 2 Report

This manuscript reports the coating of graphine with alumina to provide a material for the preparation of filled epoxy with both high thermal conductivity and low electrical conductivity.

The filled epoxy compositions are referred to as "composites" yet no evidence for composite formation is provided (simply blending a filler into epoxy does not generate a composite). These should be identified as "blends" or "filled polymers." In the abstract, the properties of an epoxy blend containing 30 wt% of the coated graphene are described without any indication of the actual composition of the filler. Later it is suggested, on the basis, of THA measurements, that the additive contains approximately 25% coating with the remainder being graphine. This needs to be clarified. [This kind of thing occurs regularly.]

The manuscript will require substantial revision for accuracy, clarity and readability. Corrections are penciled-in directly on pages of the manuscript attached. These are illustrative of the kinds of changes needed throughout. In rewriting, careful attention should be paid to the use of articles, tenses and proper sentence structure. Superfluous phrases should be avoided. Author's names and et.al. should be omitted.

The quality of the writing needs to be dramatically improved (see comments and attachment).

Author Response

Dear Reviewer,

We would like to express our sincere appreciation for your valuable feedback. We are grateful for the meticulous guidance you provided regarding the grammar and language aspects, as evidenced in the attached document.

Based on your suggestions, we have revised the description of the graphene content in the manuscript. We have corrected the statement in the abstract, clarifying that 30 wt% refers to the actual mass fraction of the composite filler or the blend. In our understanding, a composite material is composed of two or more different materials with distinct physical properties, with interfaces between the components. Our material falls under the category of filled polymers, where the composite is formed by blending different components.

Once again, we would like to express our gratitude for your guidance throughout the review process. Your input has been invaluable in improving the clarity and accuracy of our manuscript.

Reviewer 3 Report

This is an interesting manuscript that reports on the development of a multicomponent polymer composite material consisting of an epoxy matrix and alumina-coated graphene filler. This composite uniquely combines electrically insulating and thermally conductive properties. It is attributed to the deposition of alumina particles assisted by a cationic surfactant. I think that the research is novel in its approach and demonstrates a good improvement of the aforementioned material properties. Therefore I suggest its publication in Nanomaterials after minor revision.

Some questions/comments:

-        do the authors think or investigated whether the method works generally in the presence of any cationic surfactants? Possibly the amount of surfactant greatly affects the product properties.

-        “an uniformly growth of coating layer” is incorrect, please reformulate

-        what does “pyknotic” mean? If the authors prefer to use this term, they should explain it briefly

-        “adding 1g graphene” is strange, it is graphene powder and not a free-standing particle. Anyway, can the authors know there are free-standing graphene sheets in the solid powder?

-        “the use of CTAB surfactant with a positive charge is beneficial to the directional deposition of Al2O3.”: Why is that? The authors mentioned the zeta potential of graphene as negative. However, Al2O3 may be either positively or negatively charged depending on the solution pH. What was the pH of the synthesis?

The language is more or less fine, minor things are detected but not listed here.

Round 2

Reviewer 2 Report

This manuscript is much improved. A composite is a composition in which there is strong interfacial adhesion between the filler and the matrix such that the physical properties are much improved over those for the matrix alone.

Author Response

Thanks again for reviewer's guidance. The structure and language descriptions of the article have greatly improved.